# A Modified Dynamic Model of Single-Sided Linear Induction Motors Considering Longitudinal and Transversal Effects

**Hamidreza Heidari** [1,*], **Anton Rassõlkin** [1], **Arash Razzaghi** [2], **Toomas Vaimann** [1], **Ants Kallaste** [1], **Ekaterina Andriushchenko** [1], **Anouar Belahcen** [1,3] and **Dmitry V. Lukichev** [4]

1   Department of Electrical Power Engineering and Mechatronics, Tallinn University of Technology, 19086 Tallinn, Estonia; anton.rassolkin@taltech.ee (A.R.); toomas.vaimann@taltech.ee (T.V.); ants.kallaste@taltech.ee (A.K.); ekandr@taltech.ee (E.A.); anouar.belahcen@aalto.fi (A.B.)
2   Department of Electrical Engineering, University of IAU, Mianeh 5315836511, Iran; razzaghi.ieee@gmail.com
3   Department of Electrical Engineering, Aalto University, 11000 Aalto, Finland
4   Faculty of Control Systems and Robotics, ITMO University, 197101 Saint Petersburg, Russia; lukichev@itmo.ru or dmitry.v.lukichev@gmail.com
*   Correspondence: haheid@taltech.ee; Tel.: +372-56139797

**Abstract:** This paper proposes a modified dynamic equivalent circuit model for a linear induction motor considering both longitudinal end effect and transverse edge effect. The dynamic end effect (speed-dependent end effect) is based on conventional Duncan's approach. The transverse edge effect is investigated by using three correction factors applied to the secondary resistance and magnetizing inductance. Moreover, the iron saturation effect, the skin effect, and the air-gap leakage effect are incorporated into the proposed model by using the field-analysis method. A new topology of the steady-state and space-vector model of linear induction, regarding all mentioned phenomena, is presented. The parameters of this model are calculated using both field analysis and the finite-element method. The steady-state performance of the model is first validated using the finite-element method. Additionally, the dynamic performance of the proposed model is studied. The results prove that the proposed equivalent circuit model can precisely predict the dynamic and steady-state performances of the linear induction.

**Keywords:** dynamic performance; equivalent circuit model; finite-element method; linear induction motor; longitudinal end effect; transverse edge effect

## 1. Introduction

Nowadays, linear induction motors (LIMs) are widely used in industrial applications such as transportation systems, production lines of factories, electromagnetic launchers, etc. Comparing LIMs with the conventional structures to produce linear motion (including rotary electric motor and gearbox), there is no need for the mechanical interface for these types of motors, which reduces the mechanical losses and stresses. Moreover, the range of velocity and acceleration of LIMs is more extensive. However, the asymmetrical structures of LIMs in both the longitudinal and transversal directions are two main disadvantages of LIMs, which yield the longitudinal end effect and the transversal edge effect, respectively. Such phenomena lead to an increment in the complexity of the LIM modeling and control [1–3].

Dynamic and steady-state modeling of the LIM has been widely addressed in the literature. In this regard, the literature can be divided into four categories, including modified mechanical equation-based models (MMEMs) [4–6], winding function-based models (WFMs) [7–9], field theory-based models (FTMs) [10–14], and Duncan's approach based models (DAMs) [15–20].

The MMEMs are based on the fact that the final effect of the longitudinal asymmetry is producing a braking force in the opposite direction of the LIM motion. Hence, the

longitudinal end effect can be considered as a braking force in the mechanical equation of the LIM and the electrical equations of LIMs are considered as rotary induction motor (RIM) ones. The braking force due to the end effect has been modeled by using the Taylor series, which is a function of linear velocity [4,5]. In [6], the resultant propulsive force was calculated using the air-gap flux density with consideration of the longitudinal end effect. The MMEMs are simple and can also predict the dynamic performance of LIMs. However, consideration of the transverse edge effect in these models is still a challenging task.

In the WFMs, some suitable winding functions are defined for both primary and secondary parts of the LIM, and then the matrix of the LIM inductances and subsequently the terminal voltages and flux linkages are calculated. A WFM has been first introduced in [7] for a high-speed double-sided LIM (DLIM) and implemented on a single-sided LIM (SLIM) with different sets of winding functions [8,9]. The application of WFMs has received attention from researchers due to its high accuracy. The counterpart of this advantage is the complexity of WFMs because of high computation, which makes these models unsuitable for analyzing the dynamic performance of LIMs, especially in variable-speed drive systems with consideration of the transverse edge effect.

The FTMs employ N-dimensional (N = 1,2,3) field theory to obtain an accurate equivalent circuit model for the LIM while deriving its parameters. It is worth mentioning here that the majority of equivalent circuit models use field analysis because of its accuracy. However, these models usually describe the steady-state behavior of the LIM and do not give any information about the dynamic operation of the LIM. In [10], most undesirable phenomena, such as longitudinal end effects, transverse edge effect, and back-iron saturation, are considered using field analysis. In [11,12], the air-gap electromotive force (EMF) has been modified by a longitudinal end effect factor. An improved series equivalent circuit model based on field theory was presented in [13]. The secondary resistance and the magnetic inductance were modified by some coefficients to consider the longitudinal end effect and the transverse edge effect in [14].

In the DAMs, the longitudinal end effect is considered by a correction factor, as a function of linear speed, applied to the magnetizing branch of the LIM's equivalent circuit model [15]. Duncan's model is widely used for the design of variable-speed drive systems because it considers both the dynamic and steady-state performance of LIMs. In this regard, the d-q equivalent circuit model and space-vector model were presented in [16,17], respectively. A modified steady-state Duncan's model was developed in [18] to cover special phenomena such as the transverse edge effect. All aforementioned DAMs only take into account the dynamic end effect, which is related to linear speed. The dynamic d-q and steady-state equivalent circuit models with consideration of both dynamic and static end effect (or speed-independent end effect) have been investigated in [19,20], respectively.

Modeling of motors is crucial for many objectives, including life-cycle analysis, performance analysis, and more importantly, control purposes [21–23]. In this paper, a modified dynamic equivalent circuit model of LIMs is proposed. The model considers most special phenomena of LIMs, including (1) the dynamic longitudinal end effect using the conventional Duncan's approach, (2) the transverse edge effect using three correction factors for modifying secondary resistance and magnetizing inductance, (3) the iron saturation effect, (4) the skin effect and (5) the air-gap leakage effect. A new topology of the steady-state and space-vector model of LIMs is presented. The proposed model can analyze both the steady-state and dynamic performance of LIMs, and hence it is useful for obtaining an accurate variable-speed drive system. To validate the proposed model, finite-element method (FEM) is employed. The rest of the paper is organized as follows. Section 1 briefly reviews Duncan's equivalent circuit model. Section 2 describes the proposed equivalent circuit model, which includes preliminary remarks, the transverse edge effect, iron saturation, the skin effect and the air-gap leakage effect, the proposed dynamic model, and parameters' calculation of the proposed model. The results and discussion are presented in Section 3. Finally, the conclusions of the paper are synthesized in Section 4.

## 2. A Review of Duncan's Equivalent Circuit Model of LIM

The structure of a LIM is shown in Figure 1. In the LIM, when the primary part moves along with the secondary part, it continuously encounters a new material of the secondary part. Because of the appearance of this new material, the air-gap flux density is gradually increased at the entry of the primary part with a total secondary time constant that is described by $T_r = (L_m + L_{lr})/R_r$, where $L_m$, $L_{lr}$, and $R_r$ are magnetizing inductance, secondary leakage inductance, and secondary resistance, respectively. The flux density is decreased at the exit of the primary part with the secondary leakage time constant in the following way: $T_{r0} = L_{lr}/R_r$.

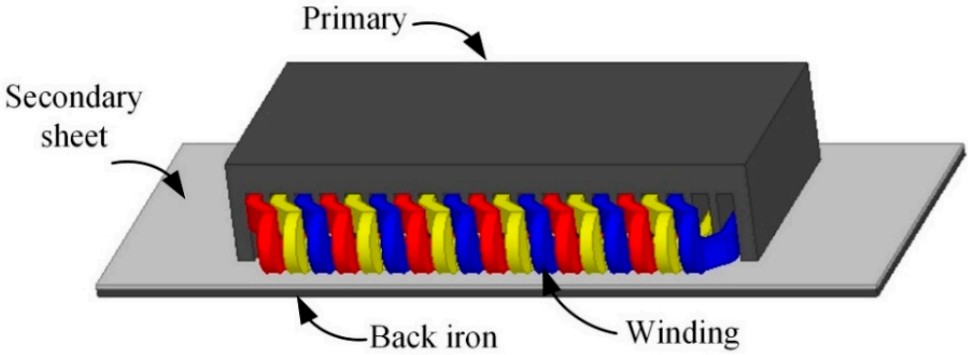

**Figure 1.** The structure of a single-sided linear induction motor (SLIM) 2. Materials and Methods.

Figure 2 shows the gradual increase and sudden decrease of the normalized air-gap flux density versus time.

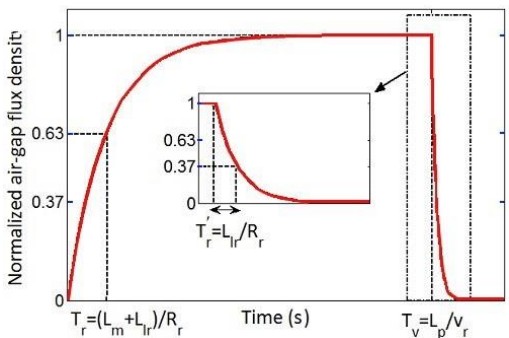

**Figure 2.** Normalized air-gap flux density versus time.

In this figure, the term $T_v = L_p/v_l$ is the time of traverse of an imaginary point by the primary core, where $L_p$ and $v_l$ are the primary lengths and linear speed, respectively. Increasing and decreasing the air-gap flux density causes an eddy current in the secondary sheet. The eddy current deteriorates the air-gap flux density in the longitudinal direction as well as increasing the ohmic losses. Such phenomena are the so-called longitudinal end effect, which can be described by the end effect factor as follows [15]:

$$Q = \frac{T_v}{T_r} = \frac{L_p/v_1}{(L_m + L_{lr})/R_r} \tag{1}$$

This factor amends the magnetizing inductance in the following way:

$$M = L_m(1 - f(Q)) \tag{2}$$

where:

$$f(Q) = \frac{1 - e^{-Q}}{Q} \tag{3}$$

## 3. Proposed Dynamic Equivalent Circuit Model of LIM

### 3.1. Preliminary Remarks

A LIM is usually made so that the widths of the primary and the secondary parts are not equal. This difference between them may lead to non-uniform distribution of the transversal flux density [24]. With the assumption that the movement direction is along the *x*-axis, the quadrature axis of that is called the *y*-axis and the transversal direction is along the *z*-axis, there is a depression in the middle area of the air-gap flux density, which has a smaller amplitude than the terminals. This phenomenon is well-known as the "transversal edge effect," which leads to an increase in the equivalent resistance of the secondary sheet. Similar to the end effect, the final influence of this phenomenon produces a braking thrust that is opposite to the developed thrust in the air-gap.

Although Duncan's model is simple and can also predict the dynamic performance of the LIM, some unwanted phenomena, particularly the transversal edge effect, are not considered in this model. This paper considers both the longitudinal end effect and the transversal edge effect. For this purpose, a dimensional structure of a LIM is illustrated in Figure 3, where $W_p$ is the primary width, $W_s$ is the secondary width, $g$ is the mechanical air-gap distance, $d_s$ is the thickness of the secondary sheet, $d_b$ is the thickness of the back iron, $h_1$ is the depth of the slot, $h_2$ is the height of the yoke, $w_1$ is the width of the primary teeth, and $w_2$ is the width of the secondary teeth. In the next sections, the modification procedure of Duncan's model will be explained.

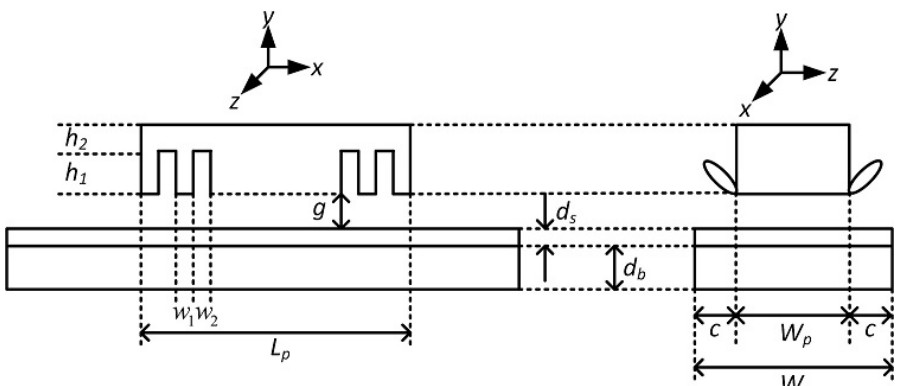

**Figure 3.** The dimensional structure of a linear induction motor (LIM).

### 3.2. Transverse Edge Effect

As mentioned earlier, the transversal edge effect results from unequal widths of the primary and the secondary parts. This effect causes an increase in the secondary resistance and a decrease in the magnetizing inductance. This paper utilizes a complex power method to consider the transversal edge effect to modify the secondary resistance and magnetizing inductance [10]. In this approach, the air-gap flux linkage is analytically calculated by Maxwell's field equations. The complex power equation is then derived by using the air-gap flux linkage. This equation is based on the structural parameters and linear velocity of the LIM. On the other hand, the complex power equation can also be achieved from an electric equivalent circuit. Thus, the coefficients of the transversal edge effect as well as the parameters of the equivalent circuit can be obtained from the equality of the analytical expressions for the complex power, which are derived from the magnetic and the electric circuits.

The transversal edge effect on the secondary sheet can be modeled using the $K_1$ and $K_2$ coefficients, which are expressed in terms of Bolton's coefficients as follows [25]:

$$K_1 = K_x \frac{1 + s^2 G^2 K_r^2 / K_x^2}{1 + s^2 G^2} \tag{4}$$

$$K_2 = \frac{K_x^2}{K_r} \frac{1 + s^2 G^2 K_r^2 / K_x^2}{1 + s^2 G^2} \tag{5}$$

where $K_r$ and $K_x$ are defined as:

$$K_r = 1 - Re\left\{(1 - jsG)\frac{2\lambda}{a\alpha}tanh(0.5a\alpha)\right\} \tag{6}$$

$$K_x = 1 - Re\left\{(Gs + j)Gs\frac{2\lambda}{a\alpha}tanh(0.5a\alpha)\right\} \tag{7}$$

The parameters that are used in the above equations are given by:

$$\lambda = \left[1 + \sqrt{1 + jsG}\,tanh(0.5a\alpha)tanh(0.5\beta(W_s - \alpha))\right]^{-1} \tag{8}$$

$$\alpha = \beta\sqrt{1 + jG} \tag{9}$$

$$\alpha = W_p + g_0 \tag{10}$$

$$g_0 = d_s + g \tag{11}$$

The goodness factor $G$ is computed as:

$$G = \frac{\omega_s \mu_0 d_s \sigma_e}{\beta^2 g_e} \tag{12}$$

In these equations, $s$ is slip, $\beta = \pi/\tau$ is the wave number, $\omega_s$ is the input frequency, $\sigma_e$ is the equivalent conductivity of the secondary part, and $g_e$ is the equivalent air-gap length. The transversal edge effect on the back iron can be expressed by inserting $\omega_s = a$ into the term of $K_2$, which is named $K_3$. As a result, the transversal edge effect on the secondary sheet and the back iron can be considered by using modification coefficients $K_1$, $K_2$, and $K_3$, which modify the magnetic inductance, secondary sheet resistance, and back iron resistance, respectively. Assume that $sG \ll 1$; these coefficients will be simplified as follows:

$$K_1 = 1 \tag{13}$$

$$K_2 = \left[1 - \frac{2tanh(0.5\alpha\beta)}{\alpha\beta[1 + tanh(0.5\alpha\beta)tanh(0.5\beta(W_s - \alpha))]}\right]^{-1} \tag{14}$$

$$K_3 = \left[1 - \frac{2tanh(0.5\alpha\beta)}{\alpha\beta}\right]^{-1} \tag{15}$$

### 3.3. Iron Saturation Effect, Skin Effect, and the Air-Gap Leakage Effect

The air-gap leakage and the iron saturation effects lead to a change in the equivalent air-gap length, which can be expressed by [10]:

$$g_e = g_0 K_1 K_c (1 + K_s) \tag{16}$$

in which:

$$K_l = \frac{sin(\beta g_0 K_c)}{\beta g_0 K_c} \tag{17}$$

$$K_s = \frac{\mu_0}{\mu_{fe}\delta_b g_0 K_c \beta^2} \tag{18}$$

where $K_c$ is Carter's coefficient, $K_l$ is the air-gap leakage coefficient, and $K_s$ is the iron saturation coefficient. $\delta_b$ is the depth of the flux density into the back iron, which is obtained as:

$$\delta_b = Re\left\{\frac{1}{(\beta^2 + j\omega_s\mu_{fe}s\sigma_b)^{0.5}}\right\} \tag{19}$$

where $\sigma_b$ is the conductivity of the back iron. The skin effect can be considered by a coefficient that modifies the equivalent conductivity of the secondary sheet as:

$$K_{sk} = \frac{d_s}{2\delta_s} \frac{\sin h\left(\frac{d_s}{\delta_s}\right) + \sin\left(\frac{d_s}{\delta_s}\right)}{\cosh\left(\frac{d_s}{\delta_s}\right) - \cos\left(\frac{d_s}{\delta_s'}\right)} \tag{20}$$

where $\sigma_s$ is the conductivity of the secondary sheet. The skin effect coefficient $K_{sk}$ modifies the equivalent conductivity of the secondary sheet as follows:

$$\sigma_{es} = \frac{\sigma_s}{K_{sk}} \tag{21}$$

Finally, the saturation effect, the skin effect, and the air-gap leakage effect lead to modification of the goodness factor $G$ (Equation (12)), in which $\sigma_e$ is equal to:

$$\sigma_e = \sigma_{es} + \frac{\delta_b}{d_s}\sigma_b \tag{22}$$

*3.4. Proposed Dynamic Model*

In the proposed equivalent circuit model of the LIM, the transversal edge effect is considered by the $K_1$, $K_2$, and $K_3$ coefficients, which modify the magnetic inductance, the resistance of the secondary sheet, and the resistance of the back iron, respectively. The longitudinal end effect is expressed using Duncan's approach. The saturation effect, the skin effect, and the air-gap leakage effect are also included by the equations that are described in Section 3.3. The proposed steady-state and dynamic equivalent circuit models are shown in Figure 4a,b, respectively.

The total secondary resistance is calculated as:

$$R_r = \frac{K_2 K_3 R_{sheet} R_{iron}}{K_2 R_{sheet} + K_3 R_{iron}} \tag{23}$$

It should be remarked that with the proposed model, both the dynamic and steady-state performances of the LIM can be analyzed. Hence, it can be used to provide an efficient variable-speed drive system for LIMs. In comparison to most WFMs or FTMs, dynamic performance prediction is the advantage of the proposed model. In comparison to DAMs, the parameters $R_r$ and $L_m$ vary with the linear velocity to consider the transverse edge effect.

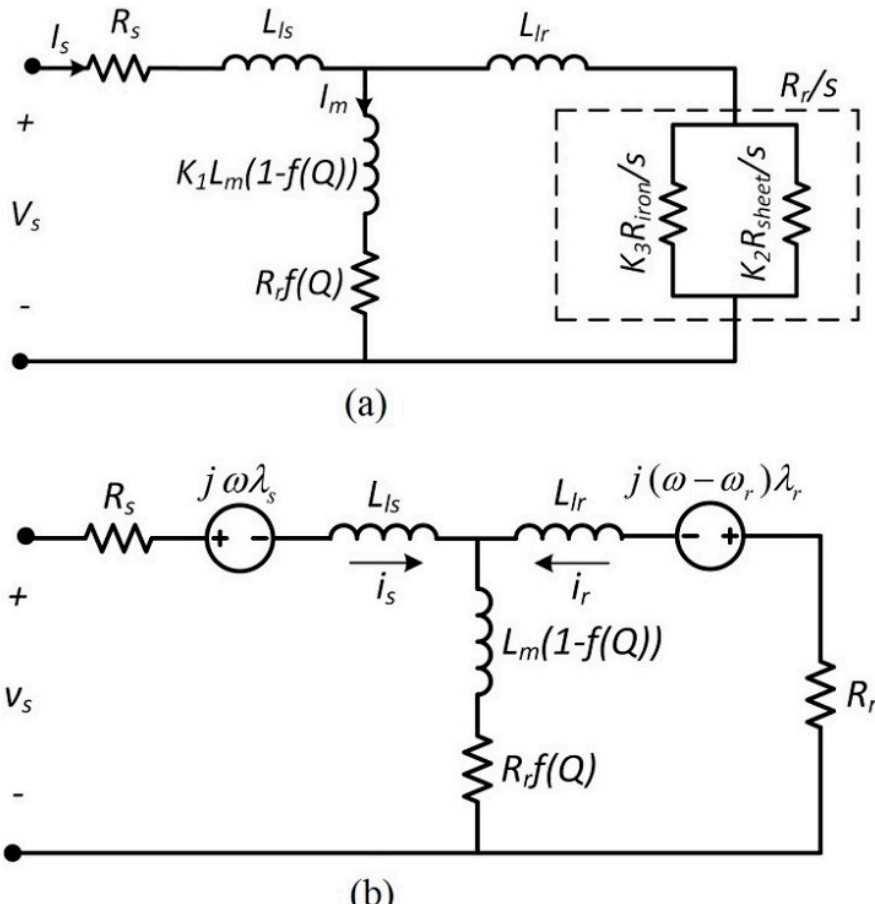

**Figure 4.** Proposed (**a**) steady-state and (**b**) dynamic equivalent circuit model of a LIM.

### 3.5. Parameters Calculation of the Proposed Model

In this paper, analytical methods based on field analysis are used for the calculation of the proposed equivalent circuit parameters, because the identification process of the equivalent circuit parameters by using practical methods, i.e., open circuit and short circuit tests, suffer from several problems so that applying these tests is impossible in some cases [26]. The resistance of the secondary sheet and the resistance of the back iron can be obtained as follows:

$$R_{sheet} = \frac{4m}{\sigma_s} \frac{(NK_\omega)^2}{p} \frac{W_p}{d_s \tau} \tag{24}$$

$$R_{iron} = \frac{4m}{\sigma_b} \frac{(NK_\omega)^2}{p} \frac{W_p}{\delta_b \tau} \tag{25}$$

where $m$ is the number of phases, $N$ is winding turns per phase, and $K_\omega$ is the winding coefficient. The magnetic inductance is calculated as [1,27]:

$$L_m = \frac{4m\mu_0 (K_\omega N)^2 (W_p + g_0) \tau}{\pi^2 p g_e} \tag{26}$$

$R_1$ is the primary resistance per phase, which is [1,27]:

$$R_1 = \frac{2(W_p + l_{ec})N}{\sigma_\omega A_\omega} \tag{27}$$

where $l_{ec}$ is the length of the end connection, and $\sigma_\omega$ and $A_\omega$ are conductivity and cross-sectional area of the primary winding conductor, respectively. The primary leakage inductance is equal to [1,27]:

$$L_{ls} = \frac{4\mu_0 N^2}{p} \left\{ \left( \lambda_s \left( 1 + \frac{3}{p} \right) + \lambda_d \right) \frac{W_p}{q} + \lambda_e l_{ec} \right\} \tag{28}$$

where $q$ is the number of slots per pole, and $\lambda_s$, $\lambda_d$, and $\lambda_e$ are permeances of slot, end connection, and air-gap leakage, respectively, which are computed as follows:

$$\lambda_s = \frac{h_1 (1 + 3K_p)}{12\omega_2} \tag{29}$$

$$\lambda_d = \frac{5g_e/\omega_2}{5 + 4g_e/\omega_2} \tag{30}$$

$$\lambda_e = 0.3(3K_p - 1) \tag{31}$$

where $K_p$ is the pitch factor.

## 4. Results and Discussion

### 4.1. Verification of Proposed Model Using FEM

To validate the proposed model, the results were compared using 3-D FEM. In this method, all undesirable phenomena that can happen in the LIM, such as a longitudinal end and transversal edge effects, are considered with acceptable accuracy. For this purpose, Ansoft/Maxwell 14.0 software was employed. The structure parameters of the LIM are tabulated in Table 1. The 3-D view of the LIM in Maxwell software is illustrated in Figure 1.

**Table 1.** Structure parameters of the LIM.

| Parameter Description | Symbol | Unit | Value |
|---|---|---|---|
| Primary frequency | $f$ | Hz | 60 |
| No. of poles | $p$ | – | 6 |
| No. of phases | $m$ | – | 3 |
| No. of slots | $z$ | – | 20 |
| No. of slots per phase per pole | $q$ | – | 1 |
| Pole pitch | $\tau$ | mm | 66.67 |
| Mechanical air-gap | $g$ | mm | 3.2 |
| Primary length | $L_p$ | mm | 400 |
| Primary width | $W_p$ | mm | 177.8 |
| Secondary sheet thickness | $d_s$ | mm | 3.2 |
| Back iron thickness | $d_b$ | mm | 6.4 |
| Secondary width | $W_s$ | mm | 247.8 |
| Slot depth | $h_1$ | mm | 52.5 |
| Yoke height | $h_2$ | mm | 26.3 |
| Opening slot | $bs_0$ | mm | 12.7 |
| Slot pitch | $\tau_s$ | mm | 19 |
| Secondary sheet conductivity | $\sigma_s$ | Ms/m | 24.59 |
| Back iron sheet conductivity | $\sigma_b$ | Ms/m | 5.8 |

Figure 5a–d show changes of saturation coefficient, goodness factor, the ratio of the equivalent conductivity to the nominal conductivity, and the ratio of the equivalent air-gap length to the nominal air-gap length with velocity, respectively. It should be mentioned that all of these figures were obtained at 60 Hz frequency. As can be seen, the fundamental parameters of the LIM were varied with a velocity that yields the changes in the electric parameters of the equivalent circuit including the secondary resistance and the magnetic inductance, while these are considered constant in conventional models.

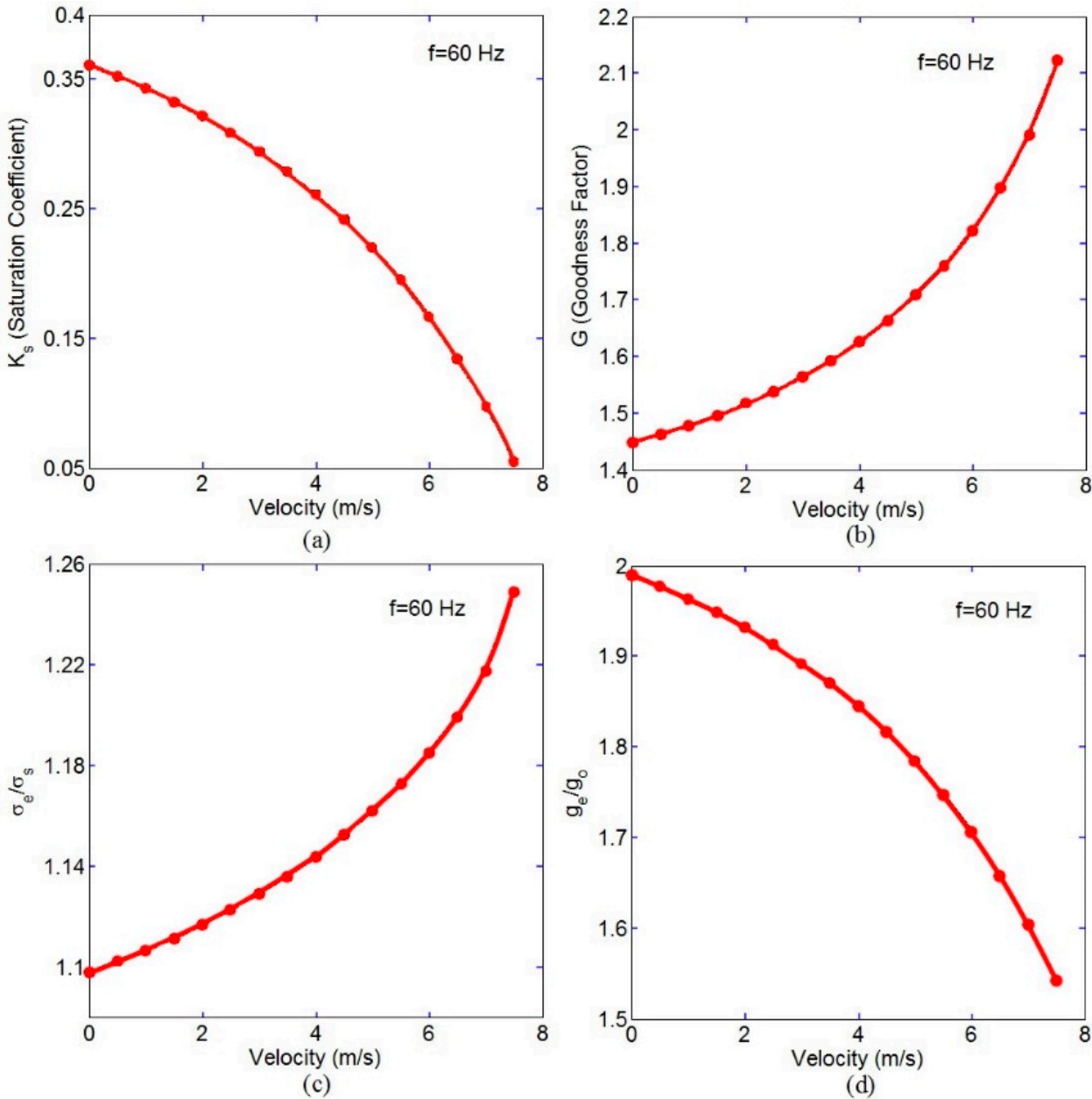

**Figure 5.** Field analysis results versus the velocity: (**a**) saturation coefficient (**b**) goodness factor (**c**) ratio of the secondary equivalent conductivity to the nominal conductivity, and (**d**) ratio of the equivalent air-gap length to the nominal air-gap length.

In this paper, the parameters of the LIM equivalent circuit are evaluated by two methods. The first one is the field analysis, which is described in Section 3.4 in detail, and the second one is FEM, which validates the results of the field analysis. Figures 6 and 7 present the magnetic inductance and the secondary resistance curves with velocity, respectively. As can be seen, the analytical method accurately provides the equivalent circuit parameters. The magnetic inductance is increased and the secondary resistance is reduced by increasing the velocity from zero to rated speed. It should be mentioned here that the longitudinal end effect is not considered in Figures 6 and 7, because it is first indicated that the changes of magnetic inductance and the secondary resistance with the velocity due to transversal edge effect, the iron effect, the skin effect, and the air-gap leakage effect. Hence, the end effect factor (Equation (1)) modifies the parallel branch of an equivalent circuit for considering the longitudinal end effect. The values of the leakage

inductance and the primary resistance from the analytical method and FEM are listed in Table 2. Figure 8 shows the thrust versus velocity characteristic of LIMs using Duncan's model, the proposed model, and FEM. As can be seen, the proposed method agrees better with the 3-D FEM. The figure shows the superiority of the proposed method against the Duncan model in all speed regions. This improvement reaches up to 10 percent in the thrust estimation at the velocity of 3.5 m/s.

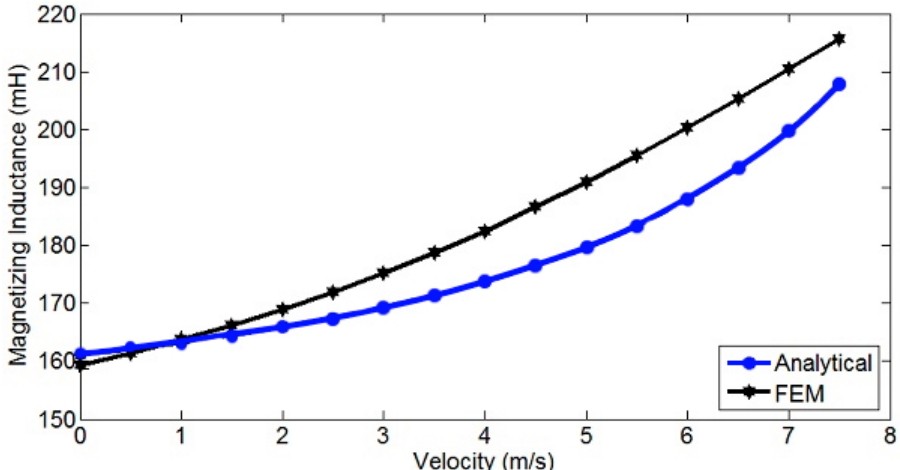

**Figure 6.** Magnetic inductance variations versus the velocity.

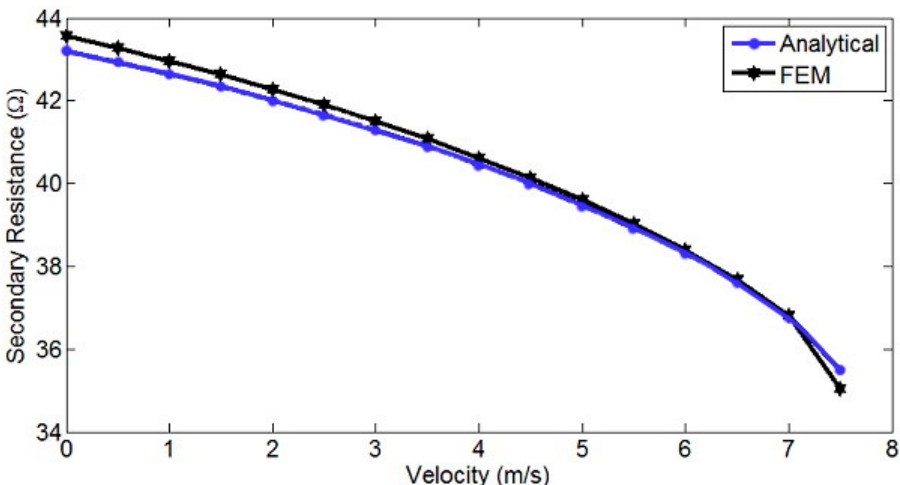

**Figure 7.** Secondary equivalent resistance variations versus the velocity.

**Table 2.** Equivalent circuit parameters of the LIM.

| Parameter | Field Analysis Method | Finite Element Method |
|---|---|---|
| $L_{ls}$ | 61.2 mH | 63.9 mH |
| $R_s$ | 10.62 Ω | 10.3 Ω |

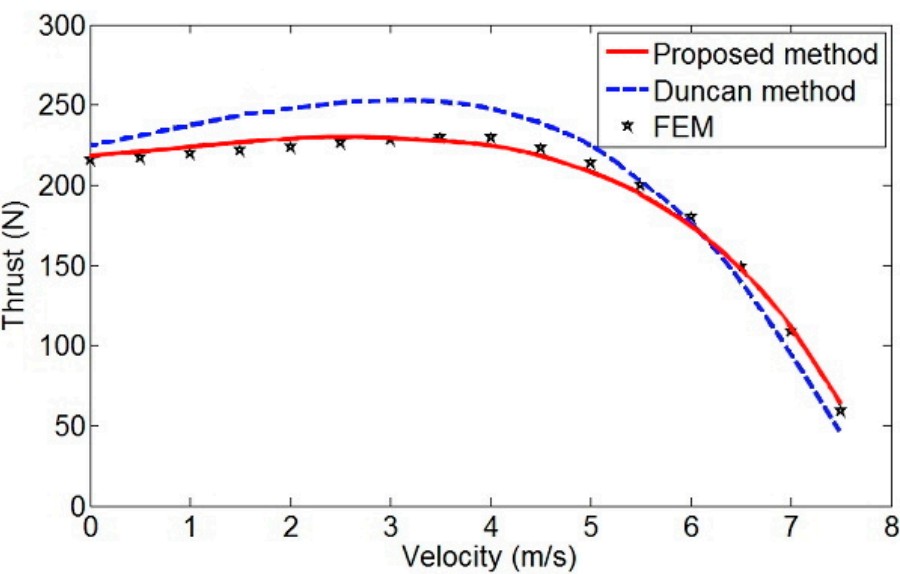

**Figure 8.** Thrust versus velocity curve.

*4.2. Dynamic Characteristics of LIMs Based on the Proposed Model*

The dynamic equivalent circuit of the LIM, in the form of the space-vector model, was shown in Figure 4b. The secondary resistance $R_r$ can be computed by Equation (23). The LIM voltage equations in the arbitrary reference frame are as follows [28]:

$$v_s = R_s i_s + j\omega\lambda_s + p\lambda_s + R_{sh}(i_s + i_r) \tag{32}$$

$$v_r = 0 = R_r i_r + j(\omega - \omega_r)\lambda_r + p\lambda_r + R_{sh}(i_s + i_r) \tag{33}$$

The flux linkages are:

$$\lambda_s = L_s i_s + M i_r \tag{34}$$

$$\lambda_r = L_r i_r + M i_s \tag{35}$$

$$L_s = L_{ls} + M \tag{36}$$

$$L_r = L_{lr} + M \tag{37}$$

$$R_{sh} = R_r f(Q) \tag{38}$$

The mechanical equation of LIMs is given as follows:

$$F_e - F_l = m_p \frac{dv_r}{dt} \tag{39}$$

where $F_e$ is the electromagnetic thrust, $F_l$ is the load force and $m_p$ is the mass of the mover. The electromagnetic thrust is calculated as:

$$F_e = \frac{3}{2}\frac{p}{2}\frac{\pi}{\tau} Re\{j\lambda_s i_s^*\} \tag{40}$$

Equations (32)–(40) are employed for dynamic performance simulation of LIMs. The values of $R_r$ and $L_m$ are acquired using look-up tables according to Figures 6 and 7. It means that the appropriate values of these parameters are determined based on LIM velocity to consider the transversal edge effect, the iron effect, the skin effect, and the air-gap leakage effect. A block diagram of the look-up tables is shown in Figure 9.

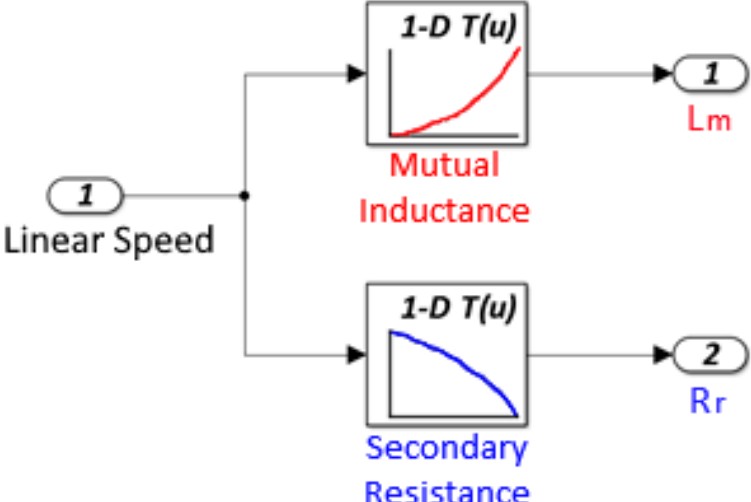

**Figure 9.** Look-up table structure for calculation of the secondary resistance and the magnetic inductance.

In the conventional Duncan's equivalent circuit (EC), the secondary resistance and the magnetic inductance are considered constant as values while they may vary with linear speed. In the proposed method, the characteristics of the secondary resistance and the magnetic inductance versus linear speed are determined using analytical methods. Then, it is used to predict the characteristic of LIMs both in transient and steady-state operation conditions. FEM is used to validate the proposed characteristic.

It is assumed that the load force Fl is equal to 50 N and the LIM is supplied by three-phase nominal voltage. Figure 10 shows the linear velocity of the LIM versus time using the proposed and Duncan's models. It is clear that the time constant of the proposed model is larger than Duncan's model, which was predictable because of greater secondary equivalent resistance at low linear velocities. Additionally, the electromagnetic thrust characteristics using the proposed model and Duncan's one are shown in Figure 11. The free acceleration characteristic of electromagnetic thrust in a LIM is similar to the free acceleration characteristic of the electromagnetic torque in a RIM. The machine accelerates to the near synchronous speed, where for running the machine, the starting thrust should be higher than the load thrust.

Firstly, this figure verifies the results obtained from Figure 8, and secondly, it demonstrates the impact of the transversal edge effect on the dynamic behavior of LIMs in comparison with Duncan's model, which only considers the longitudinal end effect.

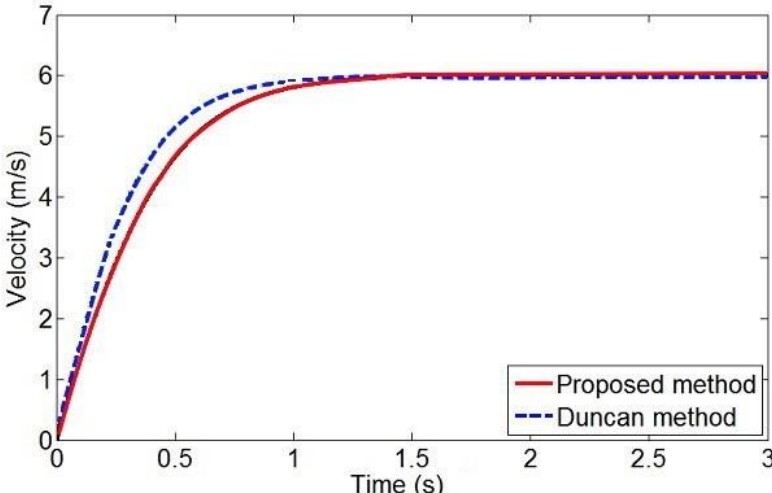

**Figure 10.** Linear velocity versus time characteristic using the proposed and Duncan's approaches.

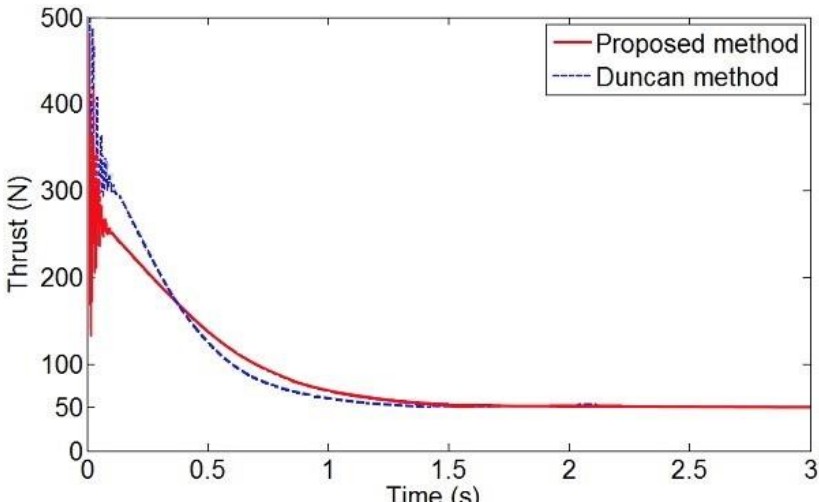

**Figure 11.** Thrust versus time characteristics using the proposed and Duncan's approaches.

## 5. Conclusions

In this paper, a new steady-state and space-vector equivalent circuit model of LIMs was proposed. For this, Duncan's model was modified so that the transversal edge effect, the iron saturation effect, the skin effect, and the air-gap leakage effect are also included alongside the longitudinal end effect. The transversal edge effect is expressed in terms of Bolton's coefficients and the other mentioned effects are incorporated in the electric parameters of the proposed equivalent circuit. The electric parameters of LIMs have been computed using both field analysis and FEM. The results show good agreement between these two approaches. To validate the proposed method, 3-D FEM was employed. Using the thrust versus velocity characteristic of LIMs, it can be derived that the proposed method provides more precision as compared to Duncan's model. In this paper, the dynamic performance of LIMs was also investigated. For this purpose, the dynamic equivalent circuit based on the proposed model was first described and then the voltage equations were provided. Values of the secondary resistance and the magnetic inductance were computed according to the linear velocity by using the look-up tables. Comparing the dynamic performance of the proposed model with Duncan's approach, it is concluded that the dynamic performance of the proposed model is slower because of consideration of the transversal edge effect and other undesirable phenomena.

**Author Contributions:** Conceptualization, methodology, validation, formal analysis, writing—original draft preparation, H.H. and A.R. (Anton Rassõlkin); writing—review and editing, resources, funding acquisition, A.R. (Arash Razzaghi); project administration, T.V.; investigation, A.K., E.A. and D.V.L.; supervision, A.B. The literature review was carried out by H.H. towards Ph.D. study under the supervision of A.R. (Anton Rassõlkin) and A.B. The study was managed by A.K., T.V., D.V.L. and E.A. collaborated in the survey for revision and collecting the database. All authors have read and agreed to the published version of the manuscript.

**Funding:** The research has been supported by the Estonian Research Council under grant PSG453 "Digital twin for propulsion drive of an autonomous electric vehicle".

**Data Availability Statement:** Data are contained within the article.

**Conflicts of Interest:** The authors declare no conflict of interest.

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
