# Peer review of "A Modified Dynamic Model of Single-Sided Linear Induction Motors Considering Longitudinal and Transversal Effects"

_electronics, doi:10.3390/electronics10080933_

Round 1

Reviewer 1 Report

  • Title normally consists of about 10-12 words, which shall not be too general or too narrow. Please consider rewording of the title to make it more attractive and eye catchier.
  • Try to avoid the usage of abbreviations in the abstract. Only last sentence of abstract represents the result of present study. Please add more information.
  • Line 28-36: Do not use sentence with bullets. Please merge them and formulate in sentence.
  • In the introduction section, after line 71, please describe the need of present study.
  • 1 & 2 should be placed under section 2. What is meaning of different colour windings in figure 1.
  • A proof reading by a native english speaker should be conducted to improve both language and organization quality.
  • Please define the different elements of figure 4. You can write the details in the caption of figure.
  • Figure 5 is placed before figure 4. Why?
  • All the figures can be more attractive. MDPI allows to publish color figures. Authors can consider it especially for figures 5-9,
  • Why authors placed figures 10 and 11 in the conclusion? Conclusion should be concise, including the most important new findings. Make sure that the conclusion is not a repetition of the abstract and only consists key conclusions.
  • Authors can also mention the limitations of the study and several recommendations on usability of this work.
  • In line 186-187, authors stated that the proposed method provides more precision as compared to the Duncan’s model. But authors did not explain the mechanism for this improvement. Can authors explain the improvement in number? Let’s say 5-10% or more.

Author Response

The authors would deeply appreciate the reviewer for the valuable comments. We believe that the comments have significantly improved the manuscript. We have scrutinized the comments and revised the manuscript based on the comments.  Please, find the responses to the comments. the revised version is also submited.

>>Title normally consists of about 10-12 words, which shall not be too general or too narrow. Please consider rewording of the title to make it more attractive and eye catchier.

The title has been modified to:

A Modified Dynamic Model of single-sided Linear Induction Motors Considering Longitudinal and transversal effects

>>Try to avoid the usage of abbreviations in the abstract. Only last sentence of abstract represents the result of present study. Please add more information.

A more detailed description of the study was presented in the abstract in the revised version.

>>Line 28-36: Do not use sentence with bullets. Please merge them and formulate in sentence.

The different modeling methods in the literature were merged and mentioned in a sentence in the revised version.

>>In the introduction section, after line 71, please describe the need of present study.

The objective of the study was added in the revised version in the mentioned section.

>>1 & 2 should be placed under section 2. What is meaning of different colour windings in figure 1.

The correction was done in the revised version. In figure 1, three-phase winding has been distinguished by different colors in the primary section.

>>A proof reading by a native english speaker should be conducted to improve both language and organization quality.

A comprehensive English and spelling edit was carried out in the revised version.

>>Please define the different elements of figure 4. You can write the details in the caption of figure.

The parameters were defined in the body text.

>>Figure 5 is placed before figure 4. Why?

The authors would appreciate the reviewer for mentioning this mistake. The figures were replaced.

>>All the figures can be more attractive. MDPI allows to publish color figures. Authors can consider it especially for figures 5-9,

The figures were changed to a colored version to make them more attractive.

>>Why authors placed figures 10 and 11 in the conclusion? Conclusion should be concise, including the most important new findings. Make sure that the conclusion is not a repetition of the abstract and only consists key conclusions.

The authors would appreciate the reviewer for mentioning this mistake. The figures were replaced.

>>Authors can also mention the limitations of the study and several recommendations on usability of this work.

In the conventional Duncan’s EC, the secondary resistance and the magnetic inductance are considered constant values while they may vary with linear speed. In the proposed method, the characteristics of secondary resistance and the magnetic inductance versus linear speed are determined using analytical methods. Then, it is used to predict the characteristic of LIM both in transient and steady-state operating conditions. FEM is used to validate the proposed characteristic. This is mentioned in the body text.

>>In line 186-187, authors stated that the proposed method provides more precision as compared to the Duncan’s model. But authors did not explain the mechanism for this improvement. Can authors explain the improvement in number? Let’s say 5-10% or more.

Please, refer to figure 8 for the comparison between the proposed model and Duncan’s model comparing FEM.

Reviewer 2 Report

This paper develops a modified dynamic equivalent circuit model of a linear induction motor  with consideration of longitudinal and transverse edge effect. For this aim, several tasks such as proposed equivalent circuit model can precisely predict the dynamic and stady-states performances  of induction LIM. Here, I have written some comments as follows: 

1-     There are some small mistakes in the Figure 3. Lp - is not primary width

2-     The authors must provide more information regarding computed values of the secondary resistance and the magnetic inductance according the linear velocity by using look-up tables .

3-     The explanation is incomplete for instance: "The electric parameters of LIM have been computed using both field analysis and FEM ." Please explain this aim in detail.

4. Please, explain in detail electromagnetic thrust characteristics using the proposed model are shown in Fig. 11.

5. In Introduction, the authors present various approaches from literature. I think that a synthesis given in a table for the solutions proposed in literature depending of the type of analysis, advantages and disadvantages, would be more useful for readers.

6. Some typing errors should be eliminated.

Author Response

The authors would appreciate the reviewer for the valuable comments. We believe that the comments have significantly improved the manuscript. We have scrutinized the comments and edited the manuscript based on the comments. Please, find the revised version and the responses to the comments. 

Here, are the comments and the responses of the authors to the comments. 

1-     There are some small mistakes in the Figure 3. Lp - is not primary width

According to Table 1, Lp is the primary length. All other parameters definition has been declared in body text and Table 1.

2-     The authors must provide more information regarding computed values of the secondary resistance and the magnetic inductance according the linear velocity by using look-up tables .

In the conventional Duncan’s EC, the secondary resistance and the magnetic inductance are considered constant values while they may vary with linear speed. In the proposed method, the characteristics of secondary resistance and the magnetic inductance versus linear speed are determined using analytical methods. Then, it is used to predict the characteristic of LIM both in transient and steady-state operating conditions. FEM is used to validate the proposed characteristic.

3-     The explanation is incomplete for instance: "The electric parameters of LIM have been computed using both field analysis and FEM ." Please explain this aim in detail.

In this paper, the field analysis is used to calculate the parameters of LIM and FEM is used to validate the results acquired from field analysis. The reason is that the results of 3-D FEM using MAXWELL software is accurate and we can assume it is near to real experimental results.

  1. Please, explain in detail electromagnetic thrust characteristics using the proposed model are shown in Fig. 11.

The free acceleration characteristic of electromagnetic thrust in LIM is similar to the free acceleration characteristic of the electromagnetic torque in RIM. The machine accelerates to the near synchronous speed, where for running the machine, the starting thrust should be higher than the load thrust.

  1. In Introduction, the authors present various approaches from literature. I think that a synthesis given in a table for the solutions proposed in literature depending of the type of analysis, advantages and disadvantages, would be more useful for readers.

The different modeling methods in the literature were merged and mentioned in a sentence in the revised version.

  1. Some typing errors should be eliminated.

A comprehensive English editing and spelling editing was carried out in the revised version.

Reviewer 3 Report

References should be updated. It is better to make experimental setup and verify the simulation results.

Author Response

The authors would appreciate the reviewer for reviewing our manuscript.

Some more updated referenced were added to the manuscript. Please, check the revised version.

The experimental setup is considered as future research work that will be studied in the future.

Best,

Round 2

Reviewer 1 Report

  • I suggested authors to avoid the usage of abbreviations in the abstract. But authors used abbreviations in the abstract and keywords.
  • I asked authors to show the improved performance (comparison with Duncan’s model) in numbers (Let’s say 5-10% or more). Authors did not add this information.
  • Graphics can be improved.

Author Response

Thank you so much for the comments.

The abbreviation was removed.

The improvement percentage was added at line 288.